# A Flexible Single Loop Setup for Water-Borne Transient Electromagnetic Sounding Applications

**DOI:** 10.3390/s21196624

**Published:** 2021-10-05

**Authors:** Lukas Aigner, Philipp Högenauer, Matthias Bücker, Adrián Flores Orozco

**Affiliations:** 1Research Unit Geophysics, Department of Geodesy and Geoinformation, Technische Universität Wien, 1040 Vienna, Austria; philipphoe@gmx.at (P.H.); adrian.flores-orozco@geo.tuwien.ac.at (A.F.O.); 2Institute of Geophysics and Extraterrestrial Physics, TU Braunschweig, 38106 Braunschweig, Germany; m.buecker@tu-bs.de

**Keywords:** geophysics, transient electromagnetic method, water-borne TEM, lake sediments, turn-off ramp, loop properties

## Abstract

Water-borne transient electromagnetic (TEM) soundings provide the means necessary to investigate the geometry and electrical properties of rocks and sediments below continental water bodies, such as rivers and lakes. Most water-borne TEM systems deploy separated magnetic transmitter and receiver loop antennas—typically in a central or offset configuration. These systems mostly require separated floating devices with rigid structures for both loop antennas. Here, we present a flexible single-loop TEM system, the light-weight design of which simplifies field procedures. Our system also facilitates the use of different geometries of the loop antenna permitting to adjust the depth of investigation (DOI) and the minimum sounding depth in the field. We measure the turn-off ramp with an oscilloscope and use the DOI to assess the minimum and maximum exploration depth of our single-loop TEM system, respectively. A reduction of the loop-antenna size improves early-time TEM data due to a reduced length of the turn-off ramp, whereas an increase of the loop-antenna size enhances the signal strength at late times, which allows to investigate deeper structures below the lake bed. We illustrate the capabilities of our system with a case study carried out at Lake Langau in Austria. Our results show that our system is capable of reaching a DOI of up to 50 m (with a maximum radius of the circular loop of 11.9 m), while it also resolves the water layer down to a minimum thickness of 6.8 m (when the radius is reduced to 6.2 m).

## 1. Introduction

Streams and lakes are important freshwater resources and often play a critical role in the recharge or contamination of groundwater. Interaction between surface- and groundwater in lakes and rivers is mainly controlled by water-bed architecture and geological composition of the corresponding aquifers and aquitards. A thorough investigation of such systems requires detailed geological information, whose acquisition through direct methods may be limited due to the challenges associated to water-borne drilling and coring. Applied geophysical methods provide a toolbox to explore the near subsurface using non-invasive techniques and provide continuous subsurface information (e.g., [1,2]). Electromagnetic methods are sensitive to the electrical conductivity of subsurface materials, which is controlled by textural properties (e.g., grain size, pore size) of soils and rocks as well as the water saturation (e.g., [3,4,5]).

The transient electromagnetic (TEM) method is well established for subsurface investigations, including hydrogeophysical investigations (e.g., [6,7]), the mapping of saline groundwater (e.g., [8,9]) as well as the assessment of the thickness of sedimentary layers in limnological studies (e.g., [10,11]). TEM applications deploy many different source-receiver combinations using mainly inductive loops (e.g., [12]). Such configurations use a transmitter loop to generate a primary magnetic field. After the transmitter current is turned off, the primary field decays, which induces a system of eddy currents in the subsurface. The secondary magnetic field of this current system contains information on the resistivity of the subsurface and can be measured using (a) the same loop (single-loop configuration), (b) a loop consisting of a parallel cable with the same shape (coincident-loop configuration), or (c) a smaller receiver loop or coil located either in the center of the transmitter (central-loop configuration) or outside of the transmitter (offset-loop configuration).

While ground-based and airborne TEM applications are standard in exploration geophysics (e.g., [13,14,15,16]), fewer studies have addressed the challenges and limitations of shallow water-borne TEM surveys. However, many recent studies have addressed the application of marine controlled-source electromagnetics (CSEM) for deep hydrocarbon exploration (e.g., [17]). Yet, in our study, we focus on the water-borne TEM method using inductive loops to characterize geologic media up to a depth of 200 m below the water table. A water-borne central loop configuration with a circular transmitter of 12.5 m radius has been used to map saline groundwater intrusions below the sea of Galilee [18]. The smaller receiver, which employs a circular multi-turn coil with a radius of 0.5 m, was transported in a second rubber boat and the system reached an approximate depth of investigation (DOI) of 60 m. Changes of river bed salinity have been mapped using a central loop configuration with a 7.5 m × 7.5 m transmitter loop and a 2.5 m × 2.5 m receiver loop [19]. The compact and rigid system reached a DOI of up to 25 m with high lateral resolution while measuring continuously and moving at speeds up to 5 km/h. A similar system has been used to investigate the sediment thickness beneath a maar lake in Germany [20]. This system uses a central loop configuration with a 18 m × 18 m transmitter and a 6 m × 6 m receiver loop. More recently, a slightly adapted version with a larger 12 m × 12 m receiver loop was used to map the hydrothermal systems of a volcanic caldera lake at the Azores [21]. TEM data were recorded both while moving continuously as well as with the boat being anchored. 1D inversion results indicate a maximum DOI of up to 140 m. The floatTEM system uses an offset loop configuration, where the 8 m × 2 m rectangular transmitter loop is towed ahead of a multi-coil receiver (0.56 m × 0.56 m) at a separation of 9 m for continuous mapping of river and/or lake bed sediments [22]. This system is based on the tTEM system developed for on-shore applications [23]. In a different application, a submerged single loop system has been used to map seafloor massive sulphides [24]. Most recently, water-borne TEM soundings with a single-loop system were used in combination with seismic, electrical resistivity and induced polarization imaging to map lake-bed sediments and investigate the sudden drying of karst lakes in Southern Mexico [25].

Besides [24,25], all of the aforementioned water-borne TEM systems use separated transmitter and receiver loops. Such systems need two different floating systems, which comprise of either two floating loops or a floating transmitter loop and an additional boat to carry the receiver. Moreover, most water-borne TEM systems require a heavy design or reinforcements to provide sufficient rigidity during the measurements, especially if the system is used for continuous measurements during navigation. In the present study, we present a cost-efficient, flexible and easy-to-build single-loop TEM system similar to the one deployed by [25] and investigate its reliability and DOI for different loop-antenna sizes. We also conduct a careful analysis of the turn-off ramp to assess early-time data quality and the possibility to characterize the electrical and geometrical properties of the lake water. The two main objectives of our study are to (1) evaluate our single-loop water-borne TEM system with respect to data quality and its capability to delineate sedimentary layers below continental fresh-water bodies, as well as (2) demonstrate the flexibility of our system, the antenna-loop size of which can be adjusted for either a high resolution of the near surface or deeper investigations.

## 2. Materials and Methods

### 2.1. Test Site

Measurements in this study were collected at Lake Langau located in the Waldviertel region in lower Austria (WGS84: $48^{\circ }50^{\prime }38.6^{\prime \prime }$ N $15^{\circ }43^{\prime }46.2^{\prime \prime }$ E, see Figure 1). Lake Langau consists of three water bodies which originate from a former open cast lignite mine that was filled with ground water in the 1960s [26]. All measurements were conducted at the largest of the water bodies (15 ha area and 15 m max. depth). Measurements with our water-borne TEM system were carried out during two campaigns at Lake Langau in Austria. In the first campaign, we compared TEM data and inversion results obtained with three different antenna sizes and measured the turn-off ramp. Additionally, we assessed the DOI and compared central-loop soundings to single-loop soundings. In the second campaign, we conducted multiple measurements at different positions to asses the structure of sediments below Lake Langau.

### 2.2. Water-Conductivity Measurements

We used an Aqua TROLL 200 data logger (manufactured by In-Situ, Fort Collins, US, [27]) to measure the variation of the electrical conductivity of the lake water ($\sigma_{w}$) with depth. The Aqua TROLL 200 is a CTD probe, which records conductivity and temperature data at a sampling rate of 2 s to obtain quasi-continuous depth profiles of electrical conductivity and temperature. The maximum depth reached during the CTD logging (when the probe touched the lake bottom) was used to determine the water depth. In the first campaign, the temperature of the lake water was constant at 4.5 ${}^{\circ }C$ and $\sigma_{w}$ was constant at ∼95 mS/m. During the second campaign, the temperature of the lake water showed a clear layering. The first layer had a conductivity of ∼120 mS/m and a temperature of 16 ${}^{\circ }C$, while the second layer had a conductivity of ∼105 mS/m and a temperature of 14 ${}^{\circ }C$.

### 2.3. Transient Electromagnetic Measurement Principle

The TEM method can be applied using a horizontal loop antenna with a cable in a closed geometry (loop), typically a circle or square. To generate a primary magnetic field, a direct current is circulated in the loop antenna. The current is then abruptly turned off, causing a change of the primary magnetic field over time, which induces eddy currents into the ground that can be described as smoke rings [28] originating from the transmitter loop and diffusing down- and outward into the subsurface (e.g., [12]). The movement and decay of the system of eddy currents generates a secondary magnetic field whose temporal change can be measured as a transient decay of voltage induced into a receiver loop antenna at the surface. If the voltages are only measured after the transmitter current is switched off, the same loop antenna can be used both as transmitter and receiver in the so-called single-loop configuration, as presented in Figure 2a. The signal strength of the secondary field depends on the electrical conductivity (σ) of the subsurface. High values of σ enhance the strength of the late-time secondary magnetic field and lead to a slower decay of the induced voltage, while low values of σ reduce the signal strength and lead to a quicker decay of the induced voltage (see Figure 2b). For details on the application of the TEM method and different configurations we refer to [29,30,31].

### 2.4. TEM Measurement System

We used a TEM-FAST 48 manufactured by AEMR (Applied Electromagnetic Research, Utrecht, the Netherlands) to collect TEM data. This system allows injecting transmitter currents of 1A or 4A and transient voltages can be measured in up to 48 logarithmically distributed time windows ranging from 4 μs to 16 ms. One sounding consists of a total number of stacked pulses given by $P_{\mathrm{tot}}=13\times n_{s}\times n_{\mathrm{as}}$, where $1\leq n_{s}\leq 20$ is the selected stack parameter and $4\leq n_{\mathrm{as}}\leq 1024$ is the number of analog stacks, which depends on the chosen duration of the transient recording. In campaign 1, we collected each sounding with 32 active time windows, which results in a time range from 4 μs to 0.9 ms and 4160 stacked transient pulses. In campaign 2, we measured each sounding with 36 active windows resulting in a time range from 4 μs to 1.9 ms and 1248 stacked pulses. We used a 24V power supply to generate a transmitter current of 4A in both campaigns. The measurements were collected with the TEM-FAST instrument being transported in a glass-fiber-reinforced plastic boat, which was also used to maneuver the loop across the lake.

### 2.5. Construction of the Floating Loop Antenna

Figure 3a illustrates the circular single-loop geometry used in this study. The antenna system consists of: (1) a copper cable, which acts as transmitting and receiving antenna, and (2) multiple PVC pipes with a diameter of 2.5 cm and a length of 3 m, which act as floating body and stabilize the circular form of the loop antenna (Figure 3b). The PVC pipes were sealed using water repellent foam and silicone and the segments where connected by sticking the tight end of one pipe into the wide end (sleeve) of the next pipe. The joint was then fixed by passing a reusable zip tie through perforations in the walls of the two pipes. This construction is shown in Figure 3b and c and results in a stiff but fast-fitting connection. Every 4 m along the loop, we attached a piece of polyethylene heating insulation to enhance the buoyancy of the construction (Figure 3d,e).

We construct the buoyant PVC ring piece by piece while pushing it directly onto the water surface with a rope connected to the first segment to finally close the loop (as presented in Figure 3d). The copper cable is directly fixed to the outside of the ring using 2 velcro fasteners (Figure 3c,e) for each segment. The total deployment time for two people is less than 45 min and the complete equipment weighs less than 20 kg. The total size of the antenna can be adjusted by changing the number of segments, which allows to adjust the DOI to the specific needs of a survey (or a part of the survey). The buoyant loop antenna is connected to the boat using ropes, which are attached at two equidistant points of a single segment. Additionally, this segment was reinforced by doubling up the connection points with 30 cm long PVC pipes. The resulting offset (*O*) to the boat increases the length of the copper cable (*l*) and the necessary total cable length is equal to l+2O. In this study, we investigate the performance of four antennas, whose parameters are shown in Table 1. Measurements in campaign 1 were obtained close to the shore and wind conditions made it necessary to tie the loop antenna at the opposing shore to keep its position fixed and avoid deformation of the loop antenna. The profile measurements carried out in campaign 2 under less windy conditions did not require anchoring.

### 2.6. Determination of the Turn-Off Ramp and the Electromotive Force

The transmitter current cannot be turned off instantaneously. Instead, it decays over a finite time, which is commonly referred to as the turn-off ramp ($t_{r}$). The duration of $t_{r}$ affects readings at early times and is therefore an essential parameter to correctly invert for the near-surface electrical conductivity ([32,33]).

The current turn-off process induces a large negative and nearly constant voltage, also referred to as electromotive force (EMF, e.g., [12]). In practice, the duration of the EMF peak is directly related to $t_{r}$ as illustrated in Figure 4c. We measure $t_{r}$ using a DSO 1084F oscilloscope manufactured by Voltcraft. In channel one, we connect a shunt-resistor with a resistance of 2.44 Ω to measure the current in the loop and, in channel two, we measure the induced EMF (see Figure 4a,b). The TEM-FAST system estimates $t_{r}$ automatically from the loop size and the injected current. Typical values range from 3 μsto 4 μs for loops with radii of 6.2 m and 11.9 m, respectively. In this study, we evaluate the validity of this estimation.

Ramp measurements were done in campaign 1 using loops with radii of 6.2 m, 9.5 m and 11.9 m on the water surface. For all loop sizes, the measuring device was located at the same off-shore position. The water depth at the center of the two largest loops was 7.4 m and 7.6 m. As the water depth was nearly constant in this area of the lake, we assume that all three measurements approximately correspond to the same 1D layered-earth subsurface with an average water depth of 7.5 m. We repeated the ramp measurements on the shore at 50 m distance from the lake using the loops with radii of 6.2 m and 9.6 m to investigate the effect of a reduced subsurface conductivity (∼100 mS/m on the lake compared to ∼50 mS/m on the shore). We also assess the effect of different loop geometries by measuring both loops with a circular (r= 6.2 m and 9.6 m) and a square shape ($sl_{\mathrm{square}}=$ 9.75 m and 15 m).

Furthermore, we compare measurements at the land position using both the single-loop and the central-loop geometry. We use a 12.5 m transmitter loop in a square geometry to generate the transient signals for both configurations. The central-loop configuration uses a 11.25 m receiver loop also in a square geometry and both loop-antennas were centered at the same point.

### 2.7. Data Processing and Inversion

Our first processing step consists in plotting all measured transients and selecting a time range ($t_{\min}$–$t_{\max}$) based on a visual assessment of the data quality. Sections with a smooth decay and the absence of negative readings indicate sufficient data quality. We also visualize the late-time apparent conductivity ($\sigma_{a}$), using the approach described in terms of the apparent resistivity by [31], which reduces the dynamic range of the curves and facilitates the detection of less obvious outliers. The three soundings of campaign 1 were filtered to individual time ranges, because each measurement was done with different loop-antenna sizes (see Table 2). In the case of campaign 2, we conducted all soundings with the same loop (r= 7.2 m) and deleted measurements outside of the range between 12 μs and 150 μs, considering that $\sigma_{a}$ readings outside of this range deviated from a smooth shape as presented in Figure 5.

After filtering, we inverted the transient data using the smoothness-constraint inversion algorithm of the ZondTEM1d software package [34]. All inversions were done using the same regularization parameter (in ZondTEM1d termed smoothing factor) of 0.01 and 30 layers in the constant initial model with the average apparent resistivity as the starting value. Additionally, all inversions where either stopped after 12 iterations or when reaching a root-mean-squared (RMS) error <0.5%. During the inversion, the thickness of the first layer was fixed to the local depth of the lake obtained from the collocated CTD measurements. Extra readings were deleted as outliers for soundings where the forward response of the inverted conductivity model did not fit the measured data. This additional filtering was only necessary for soundings L309 and L310 to obtain an adequate data fit. The final time ranges for all soundings discussed in this study are shown in Table 2.

### 2.8. Depth of Investigation and Minimum Effective Sounding Depth

To evaluate the advantages of using loops with different sizes, we calculate the depth of investigation (DOI) of selected soundings. To assess the DOI, we use two different methodologies:(a)Comparison of inversion results of the same TEM data obtained for different conductivities of the homogeneous starting model (DOI${}_{a}$ thereafter), similar to the approach proposed by [35] for electrical resistivity tomography (ERT);(b)Application of the approach by [21] that is based on the one from [36], which considers the average subsurface conductivity and the noise level of the measured data (DOI${}_{b}$ thereafter).

To obtain the DOI${}_{a}$, we compare inversion results obtained with different start models with conductivity values of 0.1 mS/m, 10 mS/m, 100 mS/m and 1000 mS/m. The DOI${}_{a}$ is then chosen to roughly separate the near-surface depth range, where the inverted conductivity values are controlled by the data, from the deeper depth range, where the inverted conductivity values are mainly controlled by their respective start model values. The DOI${}_{b}$ is given by [21]:(1)$${\mathrm{DOI}}_{b}\approx 0.55\left(\frac{M\times \bar{\rho }}{\eta }\right)$$
where *M* denotes the magnetic moment of the transmitter loop, which is equal to $I\times A_{\mathrm{Tx}}\times n$ with $A_{\mathrm{Tx}}$ being the transmitter area, *I* the injected current and *n* the number of turns of the transmitter loop. The noise level (η) is approximated by the measured voltage of the last time gate (after filtering) and $\bar{\rho }$ the average resistivity of the smooth model [21]:(2)$$\bar{\rho }=\frac{1}{{\mathrm{DOI}}_{b}}\int_{z=0}^{{\mathrm{DOI}}_{b}}\rho \left(z\right)dz$$
with ρ(z) being the inverted resistivity at a certain depth *z*. Another approach to determine the DOI is based on assessing the sensitivity of each layer by evaluating the Jacobian matrix (e.g., [37]), which offers the additional advantage of providing the sensitivity of each layer individually to identify possible ill-determined parts of the model. Such an approach was also used to evaluate the DOI in the inversion of seismic surface wave data [38]. In our study, we use only the DOI${}_{a}$ and DOI${}_{b}$ approaches as described above.

Besides the depth of investigation, we also investigate the minimum effective sounding depth $h_{\mathrm{eff}}$, which represents the shallowest depth resolved by the single-loop sounding and is given by [39]:(3)$$h_{\mathrm{eff}}=\sqrt{t_{\mathrm{eff}}\bar{\rho }},$$
with $\bar{\rho }$ being the average resistivity defined in Equation (Equation 2). The minimum effective time $t_{\mathrm{eff}}$ represents the time, after which self-induced distortions due to the length of the current switch-off ramp may be neglected at the earliest and may provide a first estimation for the filtering process. Field data may be affected by longer distortions depending on the actual shape of the loop, which leads to the necessity of removing additional early-time voltage readings. The minimum effective time can be approximated by the time after current shut-off, at which the self-transients and the impulse response of the subsurface intersect [39]. A few values for $t_{\mathrm{eff}}$ have been computed by [39] for discrete half-space resistivities and loop sizes. The authors of [39] compute $t_{\mathrm{eff}}$ for square loops. We convert these square loop side lengths to radii of circular loops of the same perimeter and interpolate between the values of $t_{\mathrm{eff}}$ of [39] to obtain values for the loops used in this study and obtain the reference chart in Figure 6.

## 3. Results

### 3.1. Quantification of the Turn-Off Ramp to Improve Early-Time Modeling

Figure 7 compares the turn-off ramps ($t_{r}$) of three loop antennas with radii of 6.2 m, 9.5 m and 11.9 m in terms of the normalized current (to 1 A) and the EMF. The direct quantification of the length of $t_{r}$ is not possible using the normalized current (see Figure 7a–c) due to oscillations in the readings distorting the turn-off process. Hence, we use the width of the first negative peak of the EMF to quantify the length of $t_{r}$ as indicated in Figure 7d–f. The length of $t_{r}$ clearly increases with increasing antenna size but is not affected by an increase of the transmitter current.

Figure 8 shows the data and inversion results for antennas with radii of 6.2 m and 11.9 m using the values of $t_{r}$ measured with the oscilloscope and the default values of $t_{r}$ implemented in the ZondTEM1D software. The data fit in terms of the root-mean-squared (RMS) error improves slightly from 0.7% ± 0.2% to 0.5% ± 0.1% when using the measured values of $t_{r}$ instead of the default values. Additionally, the comparison of the inverted conductivity of the first layer with the water conductivity ($\sigma_{w}$) measured with the CTD probe shows that both conductivity values fit much better when using the measured values of $t_{r}$, in particular when the smaller loop antenna is used. This indicates that using the measured values of $t_{r}$ improves the conductivity values close to the surface. Hence, all inversion results shown in the following sections were obtained using measured $t_{r}$ values. To calculate $t_{r}$ of the loop antenna with a radius of r= 7.2 m (1.1 μs), we use the relationship $t_{r}\left(r\right)=0.1603\;\mu s/m\;\cdot \;\;r-0.0285\;\mu s$ obtained from a linear regression analysis of the measured values (see Figure 7d–f).

### 3.2. Ramp Measurements for Different Geometries and Subsurface Conductivities

The ramp measurements show that $t_{r}$ increases from 0.96 μs to 1.87 μs when the radius of the circular loop is increased from 6.2 m to 11.9 m. The injected current does not seem to have an effect on $t_{r}$, when measuring on the highly conductive water of Lake Langau (∼100 mS/m). However, Figure 9 shows that $t_{r}$ significantly reduces, if the subsurface is more resistive, as it is the case for the measurements on land (∼50 mS/m). It is also obvious that $t_{r}$ decreases with the transmitter current, which is the expected behavior (e.g., [23]). Furthermore, a change of the antenna geometry (square or circle) does not significantly affect $t_{r}$. We observed the same effect for the loop antenna with a radius of 6.2 m (not shown here), which indicates that $t_{r}$ is mainly affected by the length of the cable in the loop antenna (i.e., loop size) and the subsurface conductivity (see Table 3). For investigations targeting the sediments underneath the lake, a detailed knowledge of $t_{r}$ may be of less importance, because deeper measurements (later times) are less affected by the turn-off ramp and the inversion results are less affected by $t_{r}$. However, investigations targeting shallow water bodies and the water column itself may require sufficiently small loops and a careful analysis of the turn-off ramp.

### 3.3. Comparison of TEM Results Using Different Antenna Sizes

We use the same experimental layout of the ramp measurements to investigate the influence of the antenna size on both the measured data and the inverted model. The observed signal level significantly increases with increasing antenna size (Figure 10a). After initial filtering to the same time range (12 μs to 150μs), we removed additional voltage readings where the model response did not fit the measured data. Especially, at late times (>120 μs), the readings of the smallest loop resulted in a poor fit in the inversion (not shown here) and were removed as noisy measurements. Accordingly, the longest transients fitted in the inversion are those corresponding to larger antennas with a higher magnetic momentum and the shortest transient is the one related to the smallest antenna size. However, the earliest useful reading that could be fitted in the inversion was measured with the smallest loop antenna, which is due to the shorter turn-off ramp. The apparent conductivity presented in Figure 10b shows that the data measured with the two largest loops align almost perfectly. The apparent conductivity measured with the smallest loop is increased by ∼20 mS/m compared to the other to curves.

The inversion results in Figure 10c show similar 4-layer models for the loops with radii of 9.5 m and 11.9 m. The second layer is more conductive (>100 mS/m) than the lake water and has a thickness of ∼12 m. The third layer starts ∼25 m below the water surface, has a thickness of 25 m and shows the lowest conductivity of 25 mS/m. The bottom layer starts ∼50 m below the water surface and shows the highest conductivity (>200 mS/m). Due to the much shorter transient, the model obtained from data measured with the smallest loop (r= 6.2 m) does not solve for the conductive fourth layer and the conductivity of the third layer is significantly decreased compared to the inverted models using data measured with the two larger loops. However, the model obtained from data measured with the smallest loop reproduces $\sigma_{w}$ (from the CTD probe) clearly better than the two larger loops, which show a misfit to $\sigma_{w}$ of almost 50.

### 3.4. Depth of Investigation and Minimum Effective Sounding Depth

The performance of loop antennas with different sizes is assessed based on the same near-shore measurements. Figure 11a–c show that the estimated DOI${}_{a}$ increases with increasing loop area. The DOI${}_{b}$ increases from 21 m to ∼46 m, when the loop area increases from 121 $m^{2}$ to 287 $m^{2}$ (see also Table 4). Such an increase of DOI${}_{b}$ is not observed when the loop area is further increased to 448 $m^{2}$. This is probably related to the similar length of the transient used in the inversion, because DOI${}_{b}$ is mainly controlled by the signal level at the last time gate (noise level η). Furthermore, DOI${}_{b}$ is only slightly affected by the conductivity of the start model and the small RMS error of all three loop antennas (0.75% ± 0.25%) shows that the solved models correctly reproduce the measured TEM data.

In general, both DOI${}_{a}$ and DOI${}_{b}$ indicate a similar variation of the interpretable depth ranges (see Figure 11a,c). However, in case of the loop antenna with a radius of 9.5 m, the DOI${}_{a}$ is approximately 10 m shallower than the DOI${}_{b}$, due to the divergence of the solved model with a starting model value of 100 mS/m from the other three models, which starts already 30 m below the water table. In this case, the DOI${}_{b}$ probably overestimates the actual depth of investigation.

For the three studied loops, $h_{\mathrm{eff}}$ increases from 6.8 m to 13.5 m with increasing loop size (Figure 11a–c and Table 4). $h_{\mathrm{eff}}$ is above the water depth only in case of the smallest loop. This is in agreement with the observation made above that only the smallest loop solves correctly for the water conductivity.

### 3.5. Comparison of the Central and Single-Loop Configuration

To properly evaluate our single-loop antenna system, we present in Figure 12 TEM data and inversion results obtained with both the single-loop and central-loop configuration (i.e., separated transmitter and receiver loops). The data measured with the central-loop configuration show distortions in the early times (4 μs to 15 μs), which are not observed in the single-loop data. Therefore, we can use a larger part of the transient for the inversion of the single-loop data than for the central-loop data. However, late-time readings show a significant increase in the apparent conductivities (see Figure 12b) of the single-loop configuration, which results in significantly increased conductivity of the bottom layer in the inversion as shown in Figure 12c. The DOI${}_{b}$ of the central-loop configuration is increased by 10 m, which is mainly related to the decreased conductivity of the bottom layer. In general, both inversion results resolve for similar contrasts in the subsurface models as shown in Figure 12c.

### 3.6. 2D Conductivity Section at Lake Langau

Figure 13 shows the 2D conductivity section obtained from the inversion of the profile measurement, which reveals four main electrical units:1The first layer corresponds to the water; it has a thickness of ∼5 m to 10 m and a conductivity of ∼90 mS/m;2The second layer has a thickness of 1 m to 5 m and a conductivity of ∼40 mS/m;3The third layer has a thickness of ∼5 m and the highest conductivity of >200 mS/m;4The bottom layer has the lowest observed conductivity <30 mS/m and extends all the way down to the depth of investigation.

The second layer is only present between 75 m and 700 m, while the third conductive layer is continuous along the entire profile. The bottom layer starts at a depth of 10 m at the first two sounding positions and its depth increases up to 20 m along the rest of the profile. The bottom layer is continuous down to a depth of 25 m, below which it contains two resistive anomalies at 75 m and 150 m and the lowest conductivity within a large anomaly at ∼600 m profile distance. The DOI${}_{b}$ (∼30 m) of the individual soundings indicates that the sensitivity of the soundings at depth is sufficient to delineate the bottom layer and the resistive anomalies.

## 4. Discussion

Our smallest single-loop configuration (r= 6.2 m; A= 121 $m^{2}$) reaches a DOI${}_{b}$ of 21 m, which is similar to the DOI reported for the system of [19]. These authors used a central-loop system with only half the transmitter loop area to investigate a river with a water conductivity of approximately 100 mS/m and a highly conductive lake bed (>200 mS/m), which is similar to the conductivity at our test sites on Lake Langau. Our two larger loop antennas (287$m^{2}$ and 448 m^2^) reach significantly lower DOIs (∼45 m) than the system of [21], which uses a central-loop configuration with a transmitter area of 324 m to reach a DOI larger than 100 m. However, these measurements were done at a lake with a water conductivity of approximately 16 mS/m and over a highly resistive lake bed (<20 mS/m). These conductivity values are significantly lower than the average conductivity at our test site, which probably explains the much larger DOI. Additionally, the system of [21] uses a larger transmitter current (up to 30A) resulting in a much larger magnetic momentum.

Most water-borne TEM studies employ a square-loop geometry (e.g., [19,20,21,22]). While we also did some initial tests with a square-loop geometry, our light-weight construction, which is based on thin PVC tubes and simple connections with zip ties, does not provide the required stability and rigidity of the floating loop. When navigating across the lakes, a corresponding earlier square loop was breaking apart frequently due to a lack of rigidity, in particular at the corners. The final circular loop proved to be much more stable and adequate for our simple and light-weight construction. Furthermore, most water-borne TEM applications employ two loop antennas to separate the transmitter and receiver, which requires two separated floating constructions or rigid structures to keep the geometry. Our proposed single-loop configuration requires only a single floating construction (i.e., PVC ring), which significantly simplifies field procedures, while still providing sufficient data quality to delineate sedimentary layers below inner-continental water bodies.

It is worth mentioning that the relatively flexible PVC ring is still being deformed during navigation from sounding station to sounding station. However, as soon as the boat reaches the next station, the stiffness of the ring is sufficient to quickly (<3 min) reestablish the circular shape of the loop before the measurement can be started. We have observed that our antenna construction is rigid enough to allow for maneuvering with the boat engine at speeds up to 5 km/h. However, when it is necessary to cover larger distances between sounding locations, our construction permits to open the loop by releasing the zip ties at the PVC segment, which are closest to the boat. In this way, the loop can easily be pulled as a line behind the boat. Dragging the opened ring also allows passing through narrower water ways (e.g., rivers connecting neighboring lakes) and relocating safely at speeds larger than 5 km/h (without breaking the PVC construction).

Furthermore, the size of our system is easily adaptable because the final size depends only on the number of connected PVC segments. In the case of smaller loops, we recommend to decrease the diameter of the used PVC tubes to increase the flexibility of the PVC segments reducing the risk of breaking a segment. For larger loops than the ones discussed here, we recommend to increase the diameter of the individual PVC segments to increase the rigidity of the loop. We have observed that loops with a radius larger than 8 m deform stronger when measuring at wind speeds greater than 20 km/h to 25 km/h, which leads to deviations from the circular shape and a reduced antenna area. Furthermore, such wind speeds also cause significant drift of the boat and the antenna depending on the actual aerodynamic drag. This may influence inversion results in the case of lateral changes of the subsurface but can easily be avoided by using an anchor, which holds the system at the exact position.

## 5. Conclusions

We have demonstrated that our single-loop antenna for water-borne TEM investigations permits measurements with different sensitivity and we believe that our approach simplifies field measurements, because our system can be deployed in less than 45 min by only 2 people. Additionally, the entire equipment weighs less than 20 kg enabling access to sites where the transportation of heavy equipment may be limited (e.g., alpine lakes). The present study evaluates our proposed single-loop water-borne TEM system at a highly conductive lake and investigates its capability to delineate sedimentary layers below continental fresh-water bodies. The DOI of the proposed system ranges from approximately 20 m to 50 m below the water table for loops with radii of 6.2 m to 11.9 m. In the case of our test site, such DOI values were sufficient to delineate the layering below Lake Langau. Furthermore, we investigate the shallowest depth of prospecting and measure the length of the current turn-off ramp ($t_{r}$) as required for an accurate resolution of the electrical properties of the lake water. Values of $t_{r}$ increase from 0.9 μs to 1.8 μs for circular loops with radii of 6.2 m to 11.9 m. Additionally, we provide a comprehensive reference chart and tables for practitioners to investigate the reliability of shallow subsurface information. We provide detailed practical information to construct and apply our single-loop system by connecting multiple PVC segments to form a floating loop. Our light-weight system components are easy to deploy at the water body resulting in simple field procedures. Therefore, our single-loop system provides a flexible and cost-efficient alternative to the more complex water-borne systems that require two separated floating systems.

## Figures and Tables

**Figure 1 sensors-21-06624-f001:**
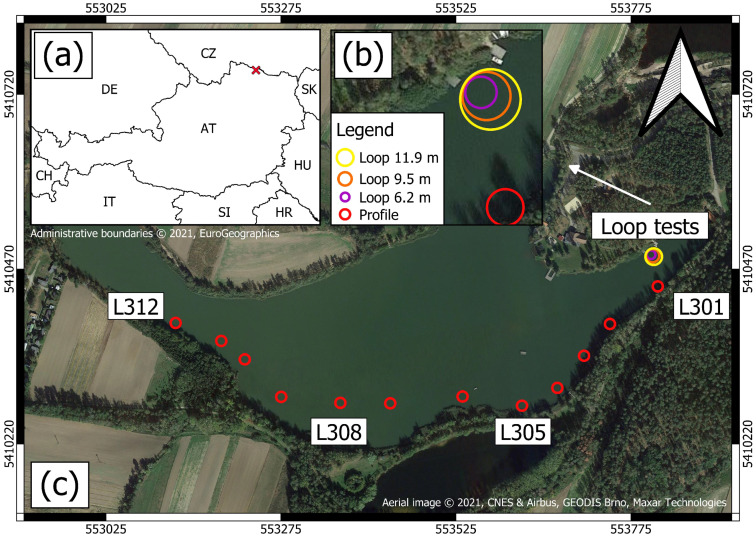
Location of the site within Austria and its neighboring countries (**a**). Aerial view of the test site to evaluate the effect of the loop size (**b**) and the measurement positions of the profile (**c**). Grid coordinate system: UTM 33N.

**Figure 2 sensors-21-06624-f002:**
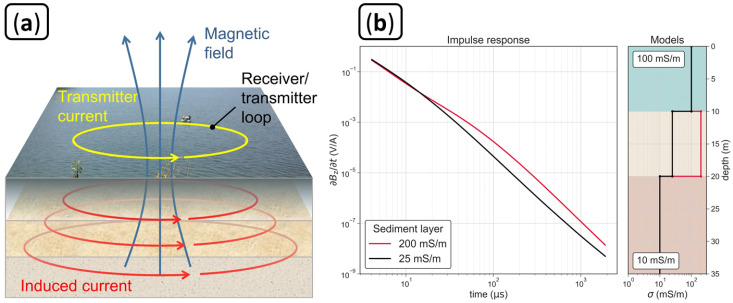
(**a**) Measurement principle of the transient electromagnetic method (TEM) for water-borne applications using a single-loop configuration, as employed in this study. (**b**) Impulse responses for two layered subsurface models with different conductivity values within the sediment layer at the lake bottom.

**Figure 3 sensors-21-06624-f003:**
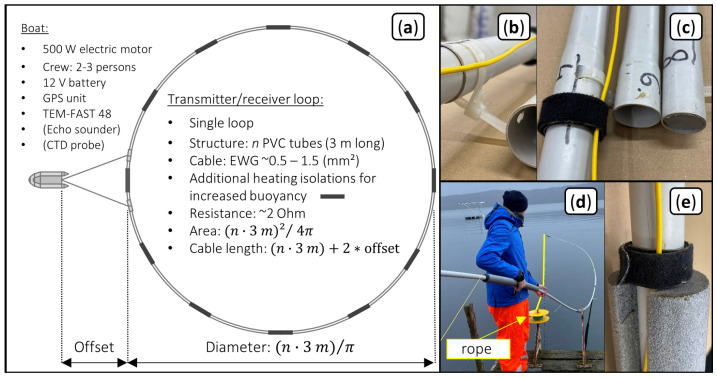
(**a**) Overview of the complete measurement system made of the loop consisting of a copper cable and a buoyant PVC ring as well as the TEM-FAST 48 system placed in the boat. (**b**) Zip tie and perforation to connect individual PVC segments. (**c**) PVC segments are sealed with water repellent foam to make the antenna float and the velcro tape is used to fasten the cable to the PVC segments. (**d**) Deployment of the PVC ring from shore using a rope to close the ring (**e**). Heating insulation is attached to the PVC pipe to increase buoyancy.

**Figure 4 sensors-21-06624-f004:**
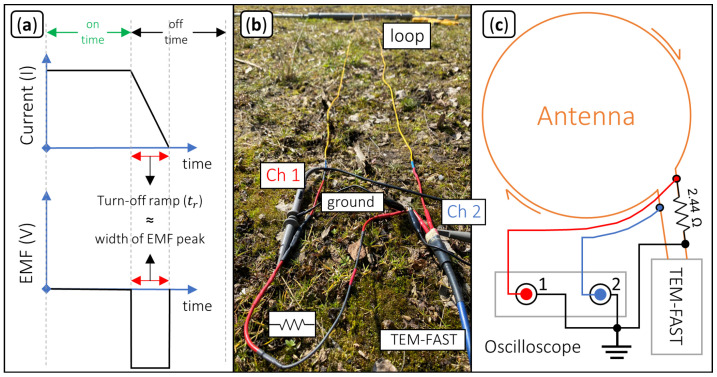
(**a**) Relationship between current and electromotive force (EMF). (**b**) Connection of the oscilloscope channels to the shunt resistor and the antenna cable for measurements of the turn-off ramp. (**c**) Circuit diagram for the determination of the turn-off ramp ($t_{r}$): Channel 1 (red) is used to measure the voltage at the shunt resistor; channel 2 (blue) is used to measure the EMF.

**Figure 5 sensors-21-06624-f005:**
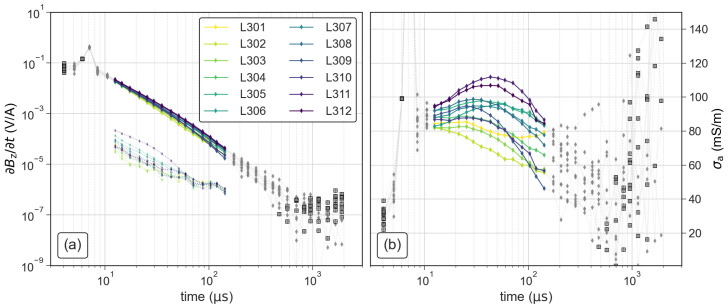
Exemplary filtering of the TEM soundings along a section measured during campaign 2 in terms of the (**a**) impulse response and (**b**) apparent conductivity. The grey symbols indicate deleted outliers, the black squares negative readings and the dotted lines in (**b**) indicate the error level at each time gate.

**Figure 6 sensors-21-06624-f006:**
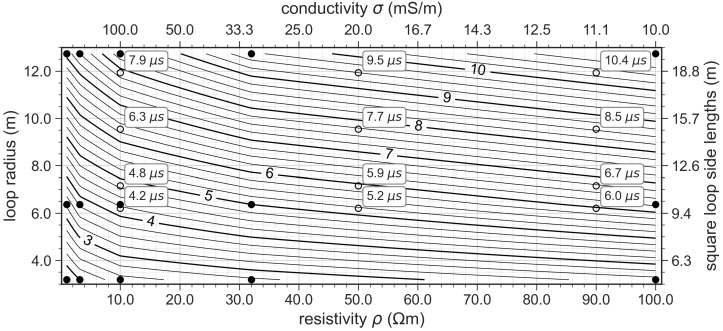
Minimum effective time $t_{\mathrm{eff}}$ (in μs) as a function of loop radius (or square-loop length) and subsurface resistivity (or conductivity). The filled circles mark the values computed by [39], which were used as input for the 2D interpolation to obtain the continuous contour lines shown here. Empty circles show the values of $t_{\mathrm{eff}}$ corresponding to our four circular loops (6.2 m, 7.2 m, 9.5 m and 11.9 m) and three different subsurface-resistivity values (10 Ωm, 50 Ωm and 90 Ωm).

**Figure 7 sensors-21-06624-f007:**
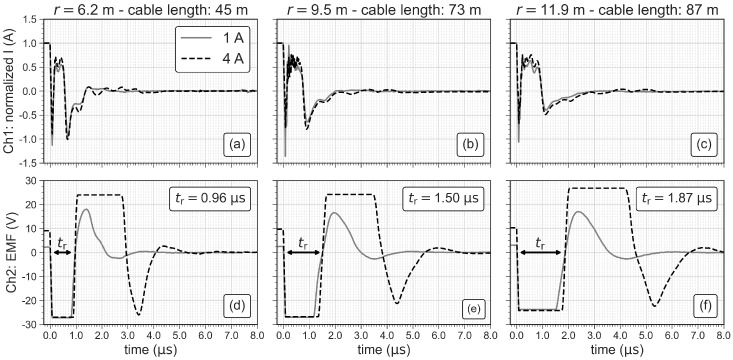
Normalized current and electromotive force (EMF) for three sizes of the loop antenna. (**a**–**c**) show the normalized current measured at the shunt resistor and (**d**–**f**) show the EMF in the antenna cables after shutting the transmitter current off.

**Figure 8 sensors-21-06624-f008:**
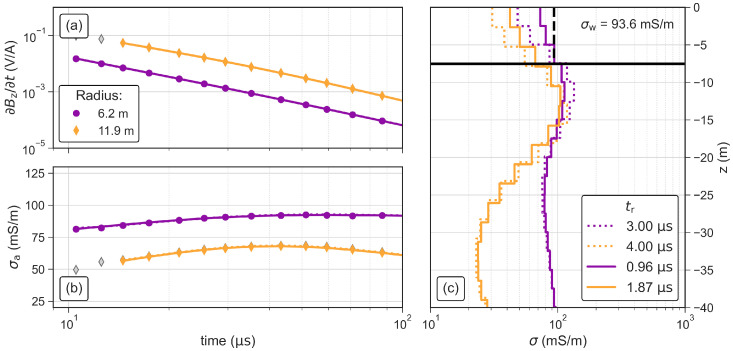
TEM soundings and inversion results obtained with circular antennas with radii of 6.2 m and 11.9 m using the default and the measured turn-off ramp times. (**a**) Impulse response and (**b**) apparent conductivity of the TEM data. Grey symbols represent readings that were removed as outliers and the solid lines indicate the model response. (**c**) Inverted models for both investigated antennas. Solid lines represent inversion results using the measured values of $t_{r}$, dotted lines the results for default values of $t_{r}$. The black horizontal line represents the water depth at the sounding position and the dashed black vertical line the water conductivity ($\sigma_{w}$) measured with the CTD probe.

**Figure 9 sensors-21-06624-f009:**
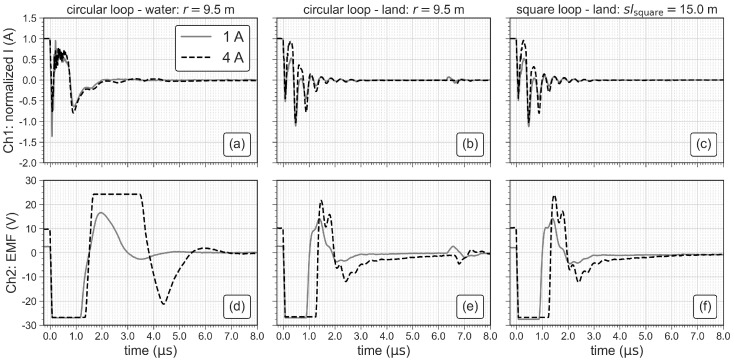
Comparison of ramp measurements with different geometries (circular and square loop), different injected currents (1A and 4A) and different average conductivities of the subsurface (∼100 mS/m on water and ∼50 mS/m on land). (**a**–**c**) show the normalized current measured at the shunt resistor and (**d**–**f**) show the induced voltage (electromotive force—EMF) in the antenna cables after shutting the transmitter current off.

**Figure 10 sensors-21-06624-f010:**
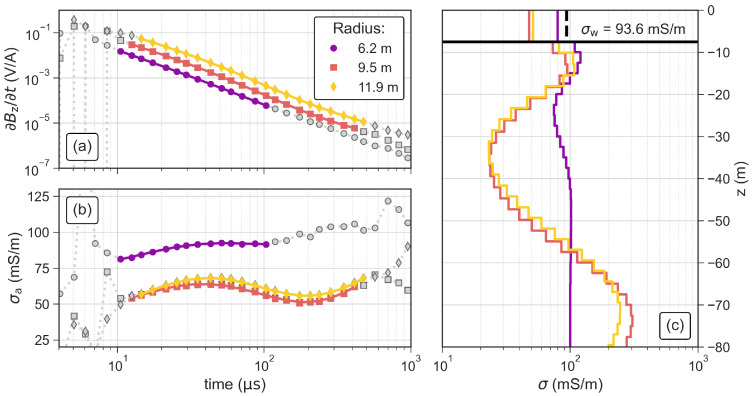
TEM soundings and inversion results obtained with circular antennas with radii of 6.2 m, 9.5 m and 11.9 m in terms of the measured and inverted (**a**) impulse response and (**b**) apparent conductivity of TEM data. Grey symbols in (**a**,**b**) represent readings that were removed as outliers and the solid lines indicate the model response. (**c**) Inverted models for each of the investigated antennas. The black horizontal line represents the lake bottom at the sounding position and the dashed black vertical line the water conductivity ($\sigma_{w}$) measured with the CTD probe.

**Figure 11 sensors-21-06624-f011:**
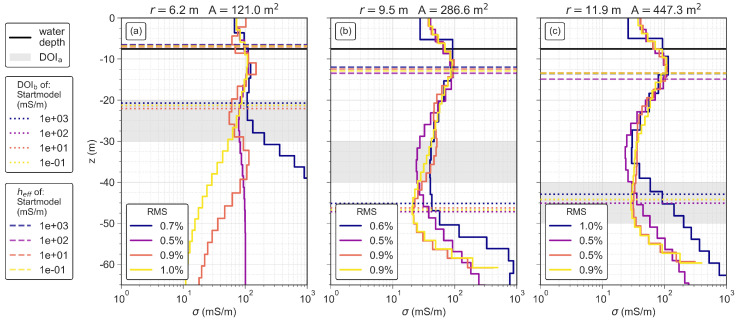
Inverted conductivity models using four different homogeneous start models for circular antennas with radii of (**a**) 6.2 m, (**b**) 9.5 m, and (**c**) 11.9 m. Each subplot shows the minimum effective sounding depth $h_{\mathrm{eff}}$, the estimated DOI${}_{a}$ range and the calculated DOI${}_{b}$ of the corresponding models. The colors of the models correspond to the respective colors of the DOI${}_{a}$ and DOI${}_{b}$.

**Figure 12 sensors-21-06624-f012:**
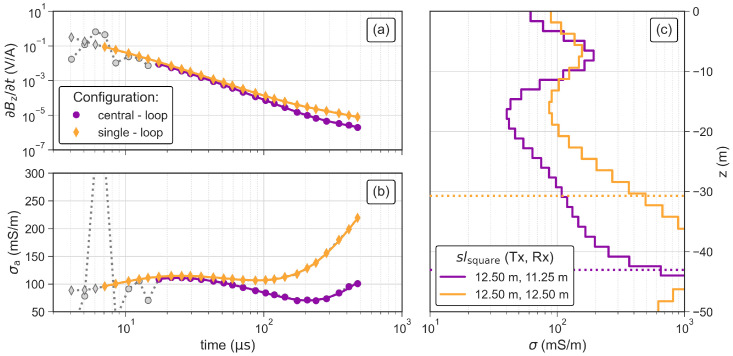
TEM soundings and inversion results obtained using the central-loop and single-loop configuration in terms of (**a**) impulse response and (**b**) apparent conductivity of the TEM data. Grey symbols in (**a**,**b**) represent readings that were removed as outliers and the solid lines indicate the model response. (**c**) Inverted models for both investigated configurations. The dotted line indicate the corresponding DOI${}_{b}$.

**Figure 13 sensors-21-06624-f013:**
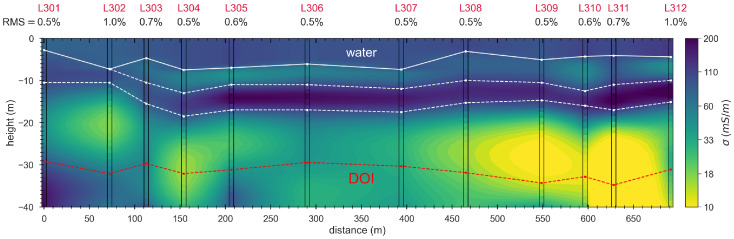
2D conductivity section obtained by interpolation of 1D inversion models at the individual sounding positions. The white line connects the water depths measured at each TEM sounding position, while the two dashed white lines highlight the approximate upper and lower limits of the third layer related to the highest conductivity (σ> 200 mS/m). The red line connects the DOI${}_{b}$ at the individual sounding locations.

**Table 1 sensors-21-06624-t001:** Characteristics of the water-borne loop antennas used in this study: $n_{\mathrm{seg}}$ is the number of PVC segments, *r* is the loop radius, *A* the loop area and *O* the offset between antenna and boat. $sl_{\mathrm{square}}$ denotes the side length of a square of the same area, *l* the length of the cable in the loop (i.e., the circumference of the circular loop), *R* is the electrical resistance and csa is the cross-sectional area of the copper wire of the loop antenna.

$n_{\mathrm{seg}}$	*r* (m)	*A* (m^2^)	*O* (m)	${\mathrm{sl}}_{\mathrm{square}}$ (m)	*l* (m)	*R* (Ω)	csa (mm^2^)
13	6.2	121.0	3.0	11.0	39	1.75	0.50
15	7.2	161.0	2.5	12.7	45	2.05	0.50
20	9.5	286.5	6.5	17.0	60	1.82	0.75
25	11.9	447.6	6.0	21.1	75	2.17	0.75

**Table 2 sensors-21-06624-t002:** Overview of TEM soundings including the sounding ID, the loop diameter *r*, the campaign, during which the sounding was done, the total number of time gates after filtering nPts, the first ($t_{\min}$) and last ($t_{\max}$) time after filtering as well as the water depth (used to constrain the first layer) at each sounding position.

ID	*r* (m)	Campaign	nPts	$t_{\min}$ (μs)	$t_{\max}$ (μs)	Water Depth (m)
L39	6.2	1	14	10.53	103.2	7.5
L60	9.5	1	21	12.55	413.8	7.5
L75	11.9	1	21	14.56	478.1	7.5
L301	7.2	2	15	12.55	142.3	2.8
L302	7.2	2	15	12.55	142.3	7.3
L303	7.2	2	15	12.55	142.3	4.7
L304	7.2	2	15	12.55	142.3	7.5
L305	7.2	2	15	12.55	142.3	7.0
L306	7.2	2	15	12.55	142.3	6.1
L307	7.2	2	15	12.55	142.3	7.4
L308	7.2	2	15	12.55	142.3	3.1
L309	7.2	2	11	14.56	70.9	5.1
L310	7.2	2	15	12.55	103.2	4.3
L311	7.2	2	15	12.55	142.3	4.1
L312	7.2	2	15	12.55	142.3	4.4

**Table 3 sensors-21-06624-t003:** Values of the turn-off ramp $t_{r}$ (in μs) measured on the land and on the water using two different cable lengths in both circular and square geometry. For all configurations and test sites, $t_{r}$ has been assessed for the two available values of the injection current (i.e., 1A and 4A).

		Circular (Water)		Circular (Land)			Square (Land)	
l **(m)**	r **(m)**	**1** **A**	**4** **A**	**1 A**	**4 A**	${\mathrm{sl}}_{\mathrm{square}}$ **(m)**	**1 A**	**4 A**
45	6.2	0.96	1.50	0.75	0.90	9.75	0.75	0.90
78	9.5	0.96	1.50	1.00	1.35	15.00	1.00	1.35

**Table 4 sensors-21-06624-t004:** Overview of the minimum effective sounding depth $h_{\mathrm{eff}}$ and the depths of investigation (DOI) for loops with different radii *r* and cross-sectional areas *A* of the circular loop. $t_{\mathrm{eff}}$ is the minimum effective sounding time. The errors of $t_{\mathrm{eff}}$ and $h_{\mathrm{eff}}$ are computed as the respective standard deviations of the four inversions using different start models. DOI${}_{a}$ represents the estimated depth range, in which the start models begins to control the inverted conductivities. The range of DOI${}_{a}$ is estimated from the visual inspection of the inverted models and is equal to the thickness of the shaded zone in Figure 11. DOI${}_{b}$ is the calculated depth of investigation; here, the errors are again computed as the standard deviations of the four inversions with different start models.

*r* (m)	A (mm^2^)	$t_{\mathrm{eff}}$ (μs)	$h_{\mathrm{eff}}$ (m)	DOI${}_{a}$ (m)	DOI${}_{b}$ (m)
6.2	121.0	4.2 ± 0.02	6.8 ± 0.2	25 ± 5	21.3 ± 0.4
9.6	286.5	7.0 ± 0.10	12.7 ± 0.6	35 ± 5	46.7 ± 0.8
11.9	447.6	8.6 ± 0.10	13.8 ± 0.6	45 ± 5	44.2 ± 0.8

## Data Availability

Data associated with this research are available and can be obtained by contacting the corresponding author.

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
