# Peer review of "A Flexible Single Loop Setup for Water-Borne Transient Electromagnetic Sounding Applications"

_sensors, 2021, doi:10.3390/s21196624_

Round 1

Reviewer 1 Report

The article "A flexible single-loop setup for water-borne transient electromagnetics" present a flexible single-loop TEM system, the light-weight design of which simplifies field procedures of water-borne transient electromagnetic (TEM) soundings. The proposed work is good, but in this reviewer’s opinion, the paper needs improvements:

1- Insert a comparison table with advantages/disadvantages between proposed method and traditional method.

2- If possible, insert a photo of the proposed setup working and the traditional setup working.

Reviewer 2 Report

The paper “A flexible single-loop setup for water-borne transient electromagnetics” discusses an interesting modification of the standard use of a commercial TDEM system (TEMFAST) for EM acquisition on water bodies.

The paper is well-organized and clearly written. However, I have some doubts about the part concerning the determination of the minimum resolvable depth and the data inversion.

The Authors can see my detailed comments on this (and other) matter in the attached pdf.

I hope this can be helpful in further improving the quality of the paper.

Regards.

Reviewer 3 Report

It is a very interesting paper, well written, with a clear description of the measurement setups and procedures. I have only minor suggestions to improve the paper quality and attractiveness:

Please, uniformize and use “water-borne”, with hyphen, as in line 1,  or “waterborne”, without hyphen, as in line 3.

----------------------------------

­Throughout the text use, for example, “2.5 m × 2.5 m”, instead “2.5 × 2.5 m”.

---------------------------------

A top view of the lake where the measurements were performed, depicting the lake contour and the measurements region ,  can make the article more attractive, and understandable, to non-European readers.

Round 2

Reviewer 2 Report

The Authors put some effort into improving the manuscript. And it has now already reached a sufficient quality level for publication.

However, in case, I still have some doubts about their replies to my previous remarks:

1)  [Original Point 2] About CSEM methods, accordingly to the Authors (and/or to my interpretation of their answer), CSEM approaches do not include inductive sources. And, for this reason, they do not need to be discussed in the present paper.

Well, CSEM methods do include inductive sources, and, given, also, the Authors’ reply, adding a few lines on this matter would be probably useful for the Reader.

2) [Original Point 3] Maybe, a reference to the In-Situ's website would be helpful.

3) [Original Point 10] The Authors’ argument that the regularization choices and inversion settings do not impact the results when applied to different datasets (due to different t_r) or that these choices equally affect the results associated with different datasets is not very convincing (if that was true I would not need to adjust, for example, the stabilizer weight, across a survey, to get comparable data fitting levels). Unfortunately, on that comparison, many of their crucial conclusions are drawn.

Having said that, I am sure the manuscript will have many Readers.
